# Human cytomegalovirus evades ZAP detection by suppressing CpG dinucleotides in the major immediate early 1 gene

Yao-Tang Lin[1]◉, Stephen Chiweshe[1]◉, Dominique McCormick[1], Anna Raper[1], Arthur Wickenhagen[2], Victor DeFillipis[3], Eleanor Gaunt[1], Peter Simmonds[4], Sam J. Wilson[2]*, Finn Grey[1]*

1 Division of Infection and Immunity, The Roslin Institute, University of Edinburgh, Easter Bush, Midlothian, United Kingdom, 2 MRC-University of Glasgow Centre for Virus Research, Glasgow, United Kingdom, 3 Vaccine and Gene Therapy Institute, Oregon Health and Science University, Portland, Oregon, United States of America, 4 Peter Medawar Building for Pathogen Research, Nuffield Department of Medicine, University of Oxford, Oxford, United Kingdom

◉ These authors contributed equally to this work.
* sam.wilson@glasgow.ac.uk (SJW); finn.grey@roslin.ed.ac.uk (FG)

**Data Availability Statement:** All relevant data are within the manuscript and its Supporting Information files.

## Abstract

The genomes of RNA and small DNA viruses of vertebrates display significant suppression of CpG dinucleotide frequencies. Artificially increasing dinucleotide frequencies results in substantial attenuation of virus replication, suggesting that these compositional changes may facilitate recognition of non-self RNA sequences. Recently, the interferon inducible protein ZAP, was identified as the host factor responsible for sensing CpG in viral RNA, through direct binding and possibly downstream targeting for degradation. Using an arrayed interferon stimulated gene expression library screen, we identified ZAPS, and its associated factor TRIM25, as inhibitors of human cytomegalovirus (HCMV) replication. Exogenous expression of ZAPS and TRIM25 significantly reduced virus replication while knockdown resulted in increased virus replication. HCMV displays a strikingly heterogeneous pattern of CpG representation with specific suppression of CpG motifs within the IE1 major immediate early transcript which is absent in subsequently expressed genes. We demonstrated that suppression of CpG dinucleotides in the IE1 gene allows evasion of inhibitory effects of ZAP. We show that acute virus replication is mutually exclusive with high levels of cellular ZAP, potentially explaining the higher levels of CpG in viral genes expressed subsequent to IE1 due to the loss of pressure from ZAP in infected cells. Finally, we show that TRIM25 regulates alternative splicing between the ZAP short and long isoforms during HCMV infection and interferon induction, with knockdown of TRIM25 resulting in decreased ZAPS and corresponding increased ZAPL expression. These results demonstrate for the first time that ZAP is a potent host restriction factor against large DNA viruses and that HCMV evades ZAP detection through suppression of CpG dinucleotides within the major immediate early 1 transcript. Furthermore, TRIM25 is required for efficient upregulation of the interferon inducible short isoform of ZAP through regulation of alternative splicing.

**Funding:** This project was funded by the Medical Research Council https://mrc.ukri.org MR/N001796/1 (FG), MR/K024752/1 (SJW), MC_UU_12014/10 (SJW) and MR/P022642/1 (SJW), the Biotechnology and Biological Sciences Research Council https://bbsrc.ukri.org, BBS/E/D/20002172 (FG), Wellcome, https://wellcome.ac.uk, WT103767MA (PS) and through the Principal's Career Development PhD scholarship from the University of Edinburgh (FG). The funders had no role in study design, data collection and analysis, decision to publish, or preparation of the manuscript.

**Competing interests:** The authors have declared that no competing interests exist.

## Author summary

The evolutionary success of viruses is dependent on their ability to circumvent defence mechanisms of the hosts they infect. These defence mechanisms rely on the ability of the host to discriminate between self and non-self, allowing identification and inhibition of invading pathogens, with the ultimate goal of containing or eradicating the infection. The specific nucleotide composition of viral genomes has been recognised as a potential target of host defences. Endogenous viruses have evolved to mirror underrepresentation of CpG motifs in mammalian genomes. Recently, the host factor ZAP has been shown to recognise and inhibit RNA viruses with artificially high CpG content, demonstrating a novel host defence mechanism. Here, we show that ZAP can potently inhibit human cytomegalovirus. In turn, the virus has evolved to reduce the levels of CpG in viral genes expressed immediately after infection of host cells, allowing evasion of ZAP recognition. This study demonstrates that in addition to targeting RNA viruses, ZAP can target large DNA viruses and, in turn, these viruses have evolved evasion mechanisms, ensuring efficient replication.

## Introduction

Interferon (IFN) is a crucial first line of defence against viral infection and shapes the adaptive immune response by triggering release of cytokines and chemokines [1, 2]. IFN expression is triggered by the recognition of pathogen-associated molecular patterns (PAMPs) [1]. These microbe-specific molecular structures are generally essential for the survival of the microbes, but fundamentally different from the host. Examples of PAMPs include peptidoglycans, lipopolysaccharide (LPS) and pathogen specific nucleic acid motifs, such as double stranded RNA and unmethylated CpG sequences within DNA. Cells recognize PAMPs through pattern recognition receptors (PRRs) that trigger innate immune responses following recognition of the target. Families of PRRs include the membrane bound Toll-like receptors, C-type lectin receptors, the cytoplasmic NOD-like receptors and RIG-I like receptors [3]. Upon recognition of the specific PAMP during invasion by a foreign pathogen, PRRs trigger signaling cascades that lead to relocation of IRF3/IRF7 complexes and NF-kB into the nucleus, initiating expression of type I IFN. In turn, activation of the IFN receptor leads to up-regulation of hundreds of IFN stimulated genes (ISGs) that, together, establish an antiviral cellular environment [4].

Dinucleotide representation in RNA sequences have been investigated as a potential PAMP [5–9]. Plant and vertebrate genomes show significantly lower CpG and TpA frequencies than would be expected given their overall base composition. Lower CpG frequencies are thought to have arisen through deamination of methylated cytosines in nuclear DNA, resulting in CpG sequences mutating to TpG [10]. RNA and small DNA viruses of vertebrates have evolved a similar pattern of suppressed CpG dinucleotides. Artificially increasing dinucleotide frequencies within their viral genomes through synonymous mutations results in considerable attenuation of virus replication [6–9].

A recent study identified the short form of the zinc-finger antiviral protein (ZAP) and its associated factor TRIM25 as responsible for recognition of high CpG frequencies in viral RNA [11]. Mammalian ZAP is expressed in two major isoforms, ZAPS (short) and ZAPL (long), which are generated by differential splicing, with ZAPL encoding an additional catalytically inactive poly (ADP-ribose) polymerase (PARP) domain [12]. ZAP had previously been identified as a host antiviral factor and is capable of binding to viral RNA through a pocket created

by the second of four zinc fingers within the RNA binding domain [13–15]. TRIM25 is an E3 ubiquitin ligase and a member of the tripartite motif (TRIM) family, many of which have been associated with antiviral functions [16, 17]. TRIM25 is required for ZAPS antiviral activity, although the precise mechanism by which TRIM25 contributes to ZAPS antiviral activity is not fully understood [18, 19]. A focused siRNA screen against human ISGs showed that knockdown of ZAP or TRIM25 rescued the replication of a defective HIV construct with artificially raised CpG levels [11]. Immunoprecipitation of ZAPS and sequencing of associated RNA demonstrated that ZAPS directly interacts with high CpG regions of HIV RNA. Knockdown of ZAPS has also been shown to rescue echovirus 7 virus that was attenuated through artificially increased CpG levels [20]. While CpG dinucleotide levels have clearly been shown to impact the fitness of RNA and small DNA viruses [6–9], their role in host recognition of large DNA viruses is less clear.

HCMV is a highly prevalent herpesvirus, persistently infecting between 30% and 100% of the human population, correlating with socio-economic status[21]. HCMV remains an important clinical pathogen accounting for more than 60% of complications associated with solid organ transplant patients [22–24]. It is also the leading cause of infectious congenital birth defects resulting from spread of the virus to neonates and has been linked to chronic inflammation and immune aging [25–27]. However, infection is normally asymptomatic due to effective control of virus replication by various arms of the immune system, including the interferon (IFN) response [2]. Although HCMV has evolved multiple mechanisms to subvert and inhibit the antiviral effects of IFNs, current evidence indicates they play a vital role in controlling replication and pathogenesis. Individuals with mutations in key IFN signalling genes are lethally susceptible to HCMV infections and recombinant IFN has been successfully used in treating congenital HCMV infection and HCMV infection in AIDS patients [28, 29]. Furthermore, murine CMV is more pathogenic in IFN knock-out mice than wildtype mice [2].

Infection with HCMV results in a robust IFN response. While viral attachment has been reported to be sufficient to trigger the IFN response, detection of HCMV is likely to occur through a multifactorial process [30]. The cellular sensors cGAS, IFI16 and ZBP1 have been shown to play a role in detection of the HCMV genome, while Toll-like receptors (TLRs) are also thought to be important [31–34]. Detection leads to activation of signalling proteins, including IRF3, IRF7 and NFKB, resulting in expression of IFN genes and subsequent up regulation of hundreds of ISGs [35–37]. Which of these ISGs are specifically responsible for inhibiting HCMV is poorly understood. While the initial ISG response is robust, it is quickly shut down by the virus, resulting in a characteristic expression profile of rapid induction at early time points (up to 24 hours post infection) followed by efficient suppression of IFN regulated gene expression. This shut down is in part due to the expression of the viral immediate early genes IE1 and IE2, both of which have been shown to curtail the initial IFN response against HCMV [35, 38, 39].

To further dissect the role of the IFN response during HCMV infection, we used an arrayed lentivirus expression library to identify ISGs that inhibit HCMV. We show that ZAP and TRIM25 can potently inhibit HCMV replication and that HCMV has evolved to suppress CpG dinucleotides in the major immediate early transcript IE1 to evade detection by ZAP.

## Results

### Arrayed ISG expression screening identified ZAPS and TRIM25 as inhibitors of HCMV replication

Systematic dissection of the 'antiviral state' using arrayed ISG expression libraries is an effective method for identifying key components of the IFN response [40–44]. To identify ISGs

that inhibit HCMV replication, human primary fibroblast cells were transduced with 421 individual ISG-encoding lentiviral vectors (or control empty vectors), in a 96-well plate format. Two days post transduction, cells were infected at a multiplicity of infection (MOI) of three with TB40/E-GFP, a BAC derived low passage HCMV strain containing an SV40 promoter driven eGFP cassette inserted between the viral TRS1 and US34 genes [45]. Levels of GFP were monitored over a seven-day period using a plate cytometer (Fig 1A). At seven days post infection, supernatant was transferred to fresh untransduced fibroblast cells to determine virus production levels. GFP levels were compared to the average signal for the respective plate. While this approach can identify which ISGs have the capacity to inhibit HCMV replication, many of the hits identified may inhibit virus replication by triggering IFN signalling through overexpression alone, independently of HCMV infection. To identify ISGs that specifically inhibit HCMV, as opposed to non-specific activation of IFN signalling, we performed a parallel screen using IRF3 KO fibroblast cells generated by CRISPR/Cas9 editing [46]. Percentage GFP levels were determined by comparing each ISG to the average GFP signal from each 96 well plate to control for inter-plate variations. Fig 1B compares the percentage virus production levels based on GFP signal for each ISG transduced well between wild type fibroblast cells and IRF3 KO cells. As virus production is the most relevant measurement for the full virus replication cycle, our studies focused on the results from this screen (The data from the primary replication screen is shown in S1A–S1C Fig and S1 Table). The blue box highlights ISGs that reduced HCMV virus production by more than 2-fold in unmodified fibroblast cells, but were not substantially inhibitory in IRF3 KO fibroblast cells. These IRF3-dependent inhibitory ISGs are highlighted in Fig 1C and include cGAS, TLR3, MyD88 and DDX60, all known to act through IRF3 signalling. Additional ISGs, not previously known to act through IRF3, showed a similar pattern. The green box in Fig 1B highlights ISGs that inhibit HCMV virus production in both wild type fibroblast cells and IRF3 KO cells, with Fig 1D showing the relative virus production levels in each cell type. These IRF3-independent ISGs include those that signal through IRF3 independent pathways (eg IRF7) or are known to specifically inhibit HCMV (IDO and RIPK2), validating the screening approach [47, 48]. Interestingly, both ZAPS and TRIM25 inhibited HCMV virus production in an IRF3 independent manner, indicating they may play a role in specifically inhibiting the virus. ZAPS and TRIM25 have recently been identified as host factors responsible for inhibiting HIV-1, Echo-7 and influenza virus constructs with synthetically increased CpG levels [9, 11, 20]. The two co-factors have also been reported to have antiviral activity against multiple viruses [49–53]. Based on this, we decided to further characterise the role of ZAPS and TRIM25 in the inhibition of HCMV replication.

A second independent screen was performed, and while there was inherent variability in the screen due to differences in lentivirus titres, the majority of top hits identified in the first screen showed a similar pattern of inhibition, including ZAPS and TRIM25 (S1 Table). However, due to this variability, it is important to carefully validate candidates of interest. To independently confirm the results of the screen, wild type fibroblast cells and IRF3 KO cells were transduced with independent lentiviral vector stocks expressing ZAPS, TRIM25, cGAS or an empty vector control, and infected with TB40/E-GFP (MOI of 3). GFP fluorescence levels were monitored for seven days (Fig 2A and 2B). As expected, cGAS inhibition is dependent on IRF3 expression, although inhibition was not completely rescued in IRF3 KO cells, suggesting cGAS may have antiviral activity that is independent of IRF3 signalling. There was also partial rescue of virus replication in IRF3 KO cells expressing ZAPS suggesting that some ZAPS inhibitory activity could depend on IRF3 signalling. However, virus replication was still significantly inhibited in wild type and IRF3 KO cells expressing ZAPS or TRIM25, corroborating the results from the screen and indicating that these ISGs inhibit HCMV virus replication in a manner independent of their role in the interferon pathway. To confirm GFP reporter levels

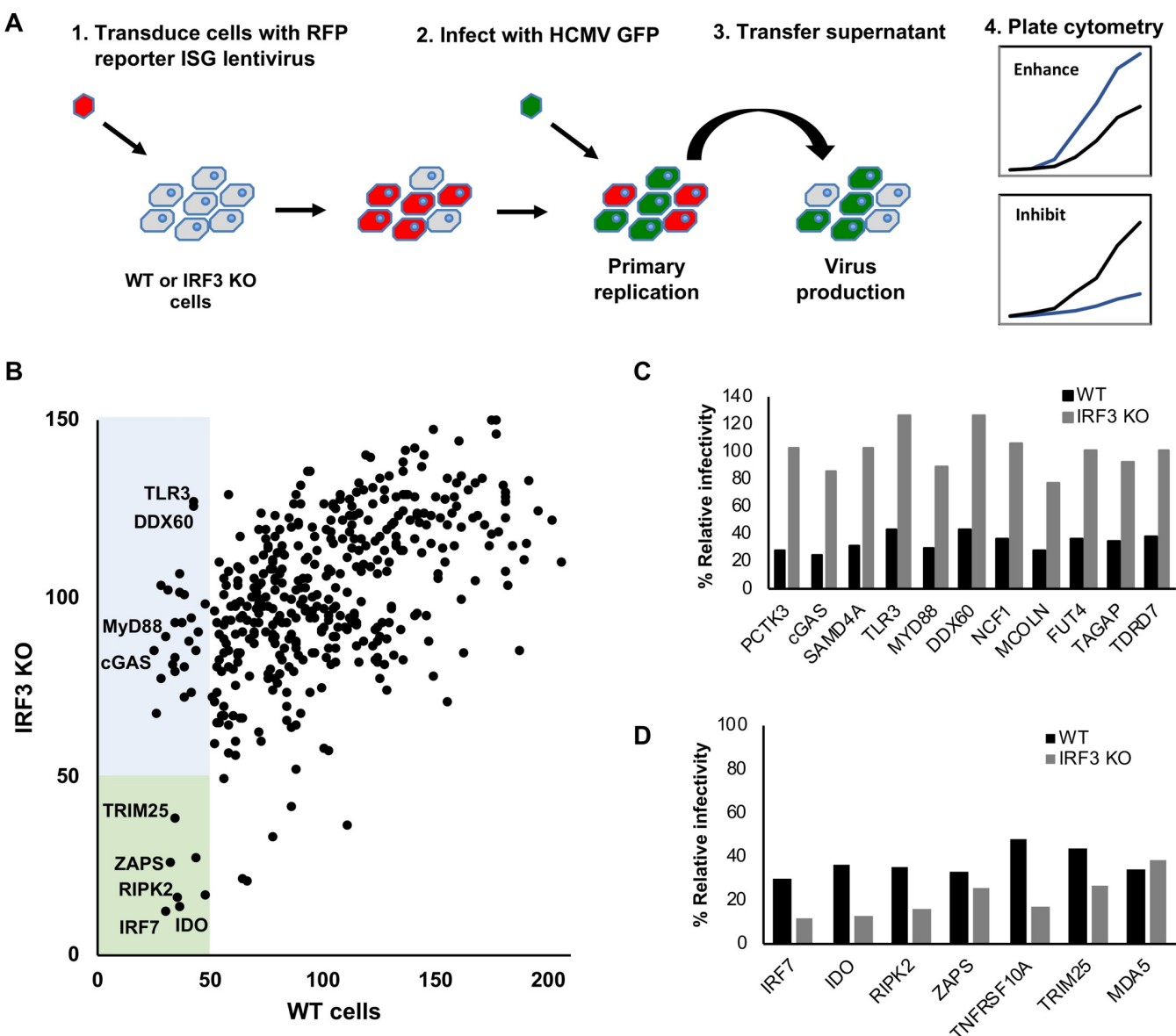

**Fig 1. Arrayed ISG expression screening identified ZAPS and TRIM25 as direct inhibitors of HCMV replication.** (A) Wild type (WT) and IRF3 knockout (KO) fibroblast cells were seeded in 96-well plates and transduced with arrayed ISG lentivirus library. Cells were infected at 48 hours post-transduction with TB40/E-GFP (MOI of 3). GFP levels were monitored over a seven-day period to measure primary replication. Seven-days post infection supernatant was transferred to untransduced cells to measure virus production (B) Direct comparison of relative HCMV virus production for each individual ISG between wild type and IRF3 KO fibroblast cells. Blue box = ISGs that reduced virus production by more than 2-fold in wild type cells, but not in IRF3 KO cells. The green box = ISGs that reduced virus production by more than 2-fold in both wild type and IRF3 KO cells. Relative primary replication and virus production levels of HCMV for the ISGs in the blue box (C) and green box (D) are shown.

expressed by the virus accurately reflected virus replication, plaque assays were performed in wild type fibroblast cells transduced with lentivirus expressing either ZAPS or TRIM25 and compared to cells transduced with an empty control lentivirus. The results confirm that expression of ZAPS and TRIM25 significantly reduced viral replication (Fig 2C). Dilution of the lentivirus input demonstrated inhibition of HCMV directly correlated with the level of ZAPS and TRIM25 overexpression (S2A and S2B Fig). As HCMV infects a range of cells *in vivo*, we also tested whether ZAPS and TRIM25 inhibited virus replication in two additional

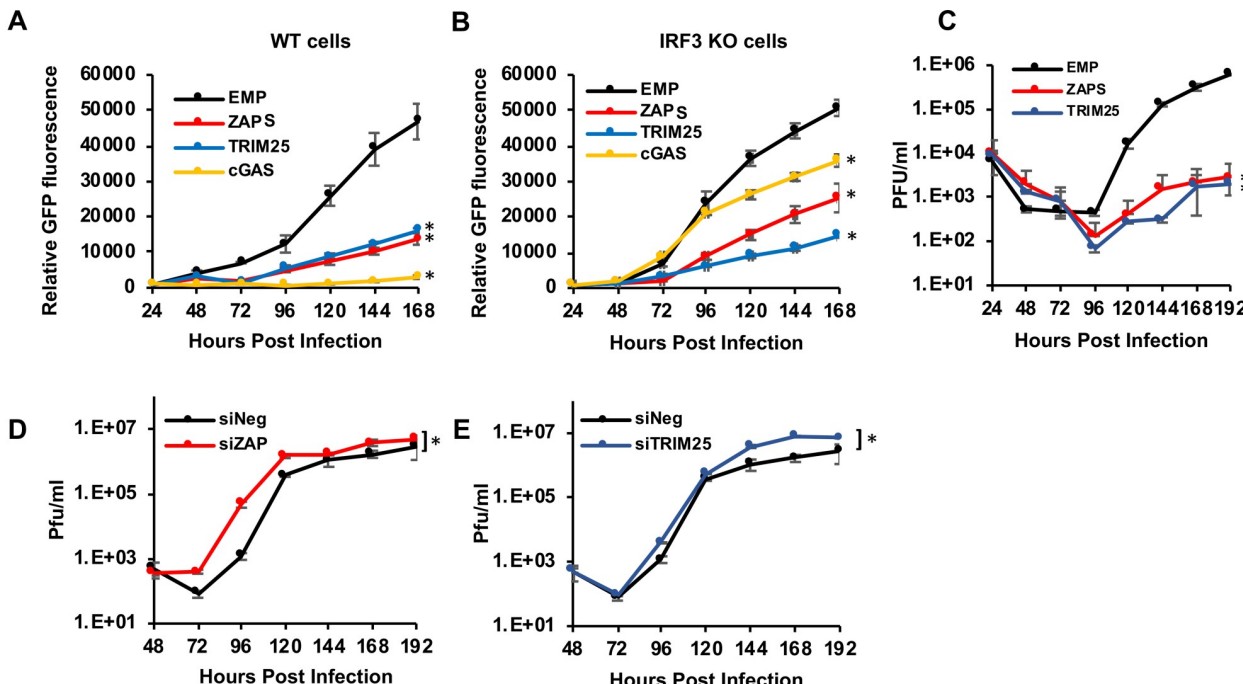

**Fig 2. ZAP and TRIM25 are host restriction factors for HCMV. Wild type cells (A) or IRF KO cells** (B) were transduced with lentivirus expressing ZAPS, TRIM25, cGAS or an empty control vector (EMP), and infected with TB40E-GFP (MOI of 3). GFP fluorescence levels were monitored for seven days by plate fluorometry. (C) Wild-type (WT) fibroblast cells were transduced with ZAPS, TRIM25 or an empty vector control (EMP), then infected with TB40E-GFP (MOI of 3). Supernatant was collected every 24 hours for 8 days, and the viral titres were determined by plaque assay. (D) WT fibroblast cells were transfected with ZAP siRNA or TRIM25 siRNA (E) and compared to cells transfected with a negative control siRNA (siNeg). 48 hours post transfection, the cells were infected with TB40E-GFP (MOI of 3). Supernatant was collected and virus titres determined by plaque assay. N = 2. * p-value < 0.05 based on two-way ANOVA.

susceptible cell types, RPE-1 (epithelial) and HUVEC (endothelial) cells. Although virus replication was less robust in these cell types, expression of ZAPS and TRIM25 significantly inhibited HCMV replication (S2C and S2D Fig). To determine the effect of knockdown of ZAP and TRIM25 on HCMV replication, fibroblast cells were transfected with siRNAs targeting ZAP, TRIM25 or a negative control siRNA. Western blot analysis demonstrated efficient knockdown of ZAP and TRIM25 (S3A and S3B Fig). Importantly, siRNA knockdown of ZAP or TRIM25 significantly increased HCMV replication, with titres increased by as much as 40-fold following knockdown of ZAP (Fig 2D and 2E). For knockdowns, pools of four siRNAs were used. To determine whether the phenotype was due to off-target effects, the siRNA pools were deconvoluted and each siRNA tested individually. In both cases, three of the four siRNAs against ZAP and TRIM25 resulted in increased virus replication, suggesting increased virus replication is not due to off-target effects (S3C and S3D Fig) Thus, endogenous ZAP can inhibit HCMV and the rescue of HCMV replication presented here is likely an underestimate, as siRNAs do not remove all the endogenous protein and ZAPS expression would be substantially higher in IFN-stimulated cells.

## Distinct patterns of dinucleotide frequencies in subfamilies of herpesvirus genomes

Recent studies have identified ZAPS and TRIM25 as host factors involved in targeting and inhibiting RNA with high CpG dinucleotide frequencies [9, 11, 20]. As such they convey evolutionary pressure on RNA and small DNA viruses, resulting in suppression of CpG

dinucleotide sequences within their genomes. However, the effects of host recognition of high CpG frequencies on larger DNA viruses, such as herpesviruses, have not been reported. Analysis of herpesvirus genomes suggests distinct patterns of CpG dinucleotide composition associated with the three subfamilies (alpha, beta and gamma) [54]. While alpha herpesviruses such as HSV-1 show no suppression of CpG dinucleotides within their genomes (S4 Fig), substantial CpG suppression is seen throughout gamma herpesvirus genomes, such as Epstein-Barr virus (EBV) (S5 Fig). Beta herpesviruses demonstrate localised suppression of CpG sequences within the major immediate early genes. These genes are the first to be expressed following infection and they are critical for driving lytic replication of the virus and are thought to play a pivotal role during the establishment, maintenance and reactivation from latency. Analysis of sixteen beta herpesvirus genomes demonstrates extensive evolutionary conservation of suppressed CpG dinucleotide sequences specifically associated with immediate early transcripts (Fig 3). In contrast, there is no such pattern for the complementary GpC dinucleotide control, (S6 Fig). These results indicate that the earliest beta herpesvirus transcripts have been selected for reduced CpG content, whereas these constraints do not extend to viral transcripts expressed at later times during infection.

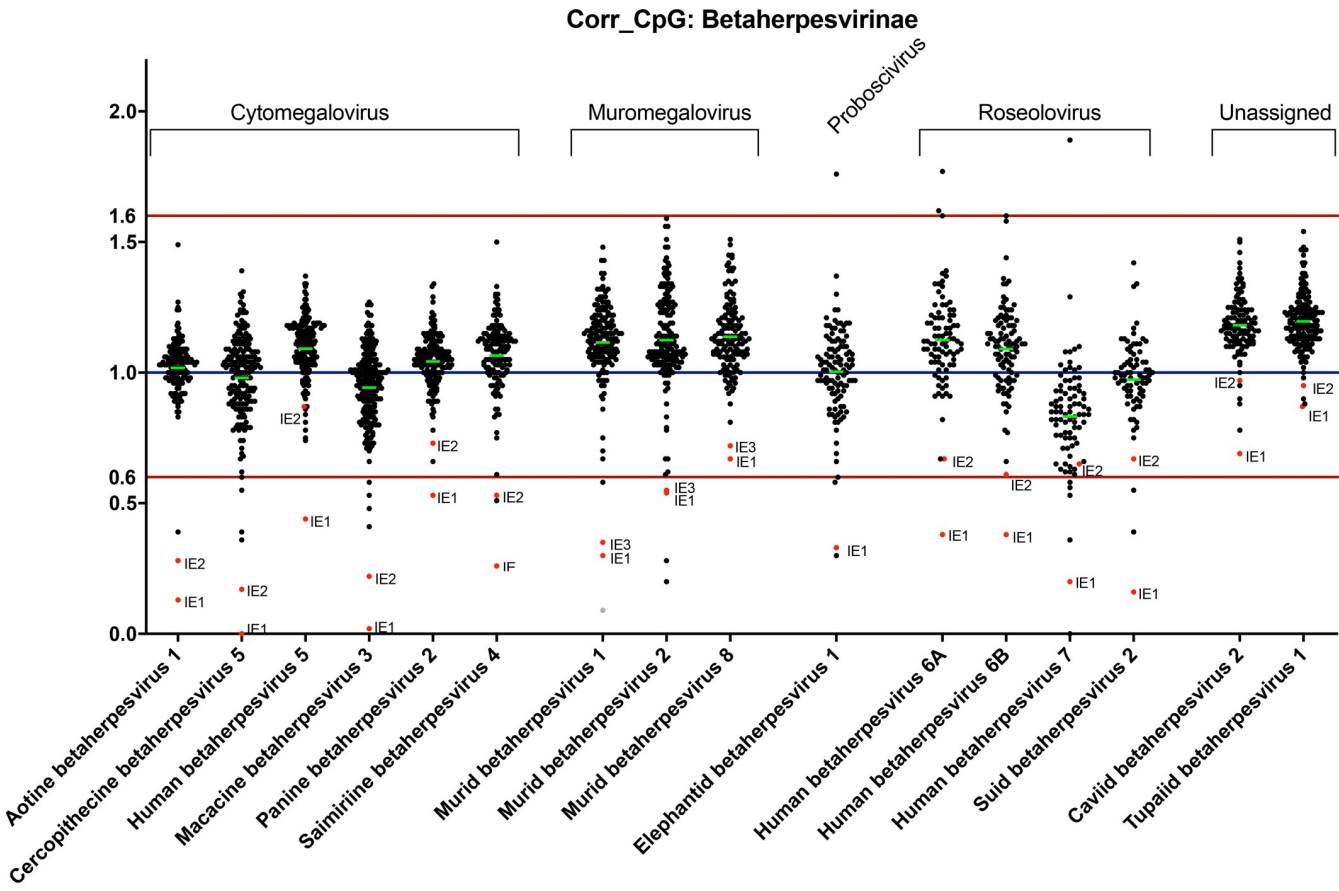

**Fig 3. Specific suppression of CpG nucleotides within the immediate early genes of beta-herpesviruses.** The CpG content of annotated open reading frames from 16 beta-herpesvirus genomes are shown, following normalization for length and GC content. A corrected CpG ratio of one reflects the expected number of CpGs based on GC content of a transcript.

## IE1 expression is not affected by ZAP

To determine the effect of ZAP expression on viral genes, the open reading frames of IE1, IE2 and pp52, an early gene with a high CpG dinucleotide frequency, were cloned into expression vectors. Cellular GAPDH tagged with T7, that has a low CpG frequency in line with mammalian genomes, was cloned into the same expression vector to generate a negative control. These plasmids were co-transfected with a plasmid expressing ZAPS. Expression of ZAPS had little effect on T7-GAPDH (Fig 4A). Similarly, while expression of ZAPS profoundly reduced pp52 expression by approximately 80% (Fig 4B) and IE2 expression to a lesser extent, ZAPS had no effect on IE1 expression. These results are in line with what would be predicted based on the CpG dinucleotide frequencies and confirm that suppression of CpG levels in IE1 allow this gene to evade the inhibitory effects of ZAPS. Interestingly, overexpression of ZAPL shows the same effects as ZAPS, suggesting that the longer isoform has the same ability to reduce expression of transcripts with high frequencies of CpG sequences (Fig 4C and 4D). A dilution series of the ZAPS vector demonstrated that increased expression of ZAPS resulted in an inverse correlation with pp52 expression levels, demonstrating the specific effect of ZAPS on pp52 expression levels (Fig 4E). Previous reports have suggested that ZAP may function through degradation of target transcripts, while other reports suggest ZAP affects translation [55, 56]. To determine whether inhibition of IE2 and pp52 were occurring at the transcript level, RT-qPCR was performed. Total RNA was harvested from the same samples used for the western blot experiments to ensure direct comparison between protein and RNA levels. Strikingly, no reduction in transcript levels was observed following expression of ZAPS or ZAPL (Fig 4F). Surprisingly, increased total transcript levels were observed for all three viral genes, although the increase in pp52 transcript levels was moderate. The larger increase in IE2 transcript levels may be an artefact related to the ability of IE2 to inhibit the CMV MIE promoter, which is part of the pcDNA vector that GAPDH-T7 and the viral genes were cloned into [57]. GAPDH and pp52 are not thought to affect the CMV promoter which may explain why similar increases in RNA levels were not observed following expression of ZAPS and ZAPL. However, the data clearly shows that reduction in pp52 protein levels caused by ZAP expression is not due to mRNA degradation, indicating a post transcriptional mechanism of inhibition.

## Artificially increasing CpG levels in IE1 results in ZAP inhibition

To determine whether the inability of ZAP to inhibit IE1 expression is due to supressed CpG levels, synonymous mutations were introduced into the IE1 coding region to artificially increase the CpG frequency (S7 Fig). As predicted, western blot analysis demonstrated that co-transfection with ZAPS or ZAPL expressing plasmid resulted in decreased expression of high CpG IE1, compared to a control vector (Fig 5A and 5B). This data confirms that reduced CpG content protects IE1 from targeting by ZAP.

## IE1 expression is not affected by ZAP during HCMV infection

As previously shown, overexpression of ZAPS profoundly inhibited virus replication (Fig 2A–2C), and overexpression of ZAPS and ZAPL resulted in the reduced expression of HCMV viral proteins encoded by genes with high CpG frequencies following transient transfection experiments in HEK293T cells (Fig 4). To determine the effect of ZAPS on viral gene expression, western blot analysis was performed on total protein samples from wild type fibroblast cells transduced with lentivirus expressing ZAPS and infected two days later with TB40E-GFP (MOI of 3) (Fig 6A). As is the case with all herpesviruses, viral gene expression occurs in a controlled temporal cascade with immediate early (IE) early (E) and late (L) gene expression occurring in a sequential fashion. Total protein was harvested at the indicated time points and

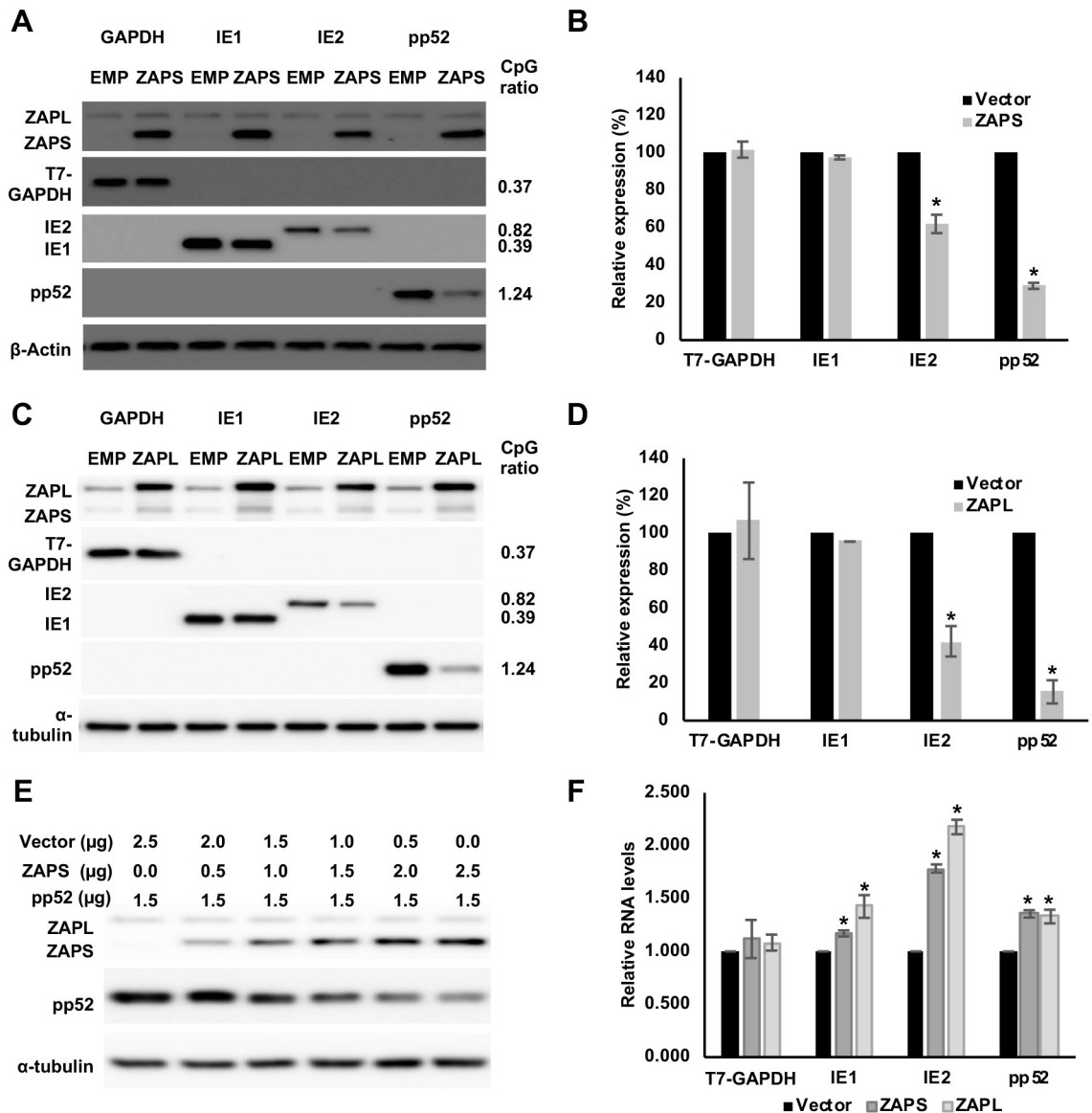

**Fig 4. ZAP targets HCMV genes with high CpG dinucleotide frequencies.** (A) 293T cells were co-transfected with plasmids expressing ZAPS (A), ZAPL (C) or a control empty vector (EMP) with plasmids expressing HCMV genes with varied CpG dinucleotide frequencies (IE1, IE2 and pp52). A plasmid expressing T7-tagged GAPDH was included as a control with low CpG sequence content. CpG frequencies are indicated beside each gel. Expression levels of HCMV genes and T7-GAPDH were determined by western blot analysis. Expression levels for proteins co-expressed with ZAPS (B) and ZAPL (D) were quantified using Image J software and normalized to the empty construct control samples (N = 2). (E) 293T cells were transfected with plasmids expressing ZAPS and pp52 at concentrations shown. Total concentration of transfected DNA was normalized using empty vector. Western blot analysis shows pp52 expression levels versus ZAP expression levels. (F) Total RNA was harvested from the same samples used for western blot analysis in (A) and (C) with transcript levels determined by RT-qPCR. N = 2, * p-value < 0.05 based on Student's t-test.

levels of IE (IE1 and IE2), E (pp52) and L (pp28) proteins measured. Strikingly, ZAPS had little to no effect on IE1 expression levels throughout the time course, but substantially reduced IE2 levels and subsequent E and L viral gene expression. In contrast, knockdown of ZAP by siRNA resulted in increased expression of pp52 and pp28 (Fig 6B). As IE2 was not excessively reduced by ZAP expression following co-transfection experiments, we do not believe the reduction in IE2 expression observation in Fig 6A is a direct result of ZAPS inhibition, but rather ZAPS

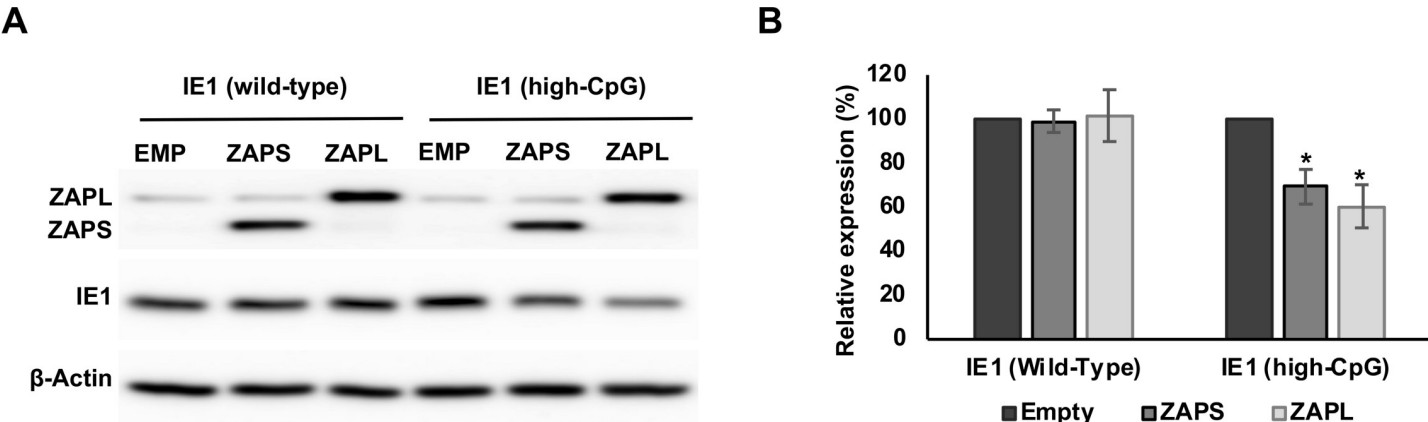

**Fig 5. Increasing CpG content in IE1 results in inhibition by ZAP.** The CpG content of IE1 was artificially increased by introducing synonymous mutations into exon 4 of the gene and cloned into an expression vector. 293T cells were co-transfected with plasmids expressing ZAPS or ZAPL or an empty control vector (EMP) and total protein harvested. (A) Western blot analysis shows levels of IE1 WT versus IE1 high CpG following expression of ZAPS or ZAPL. (B) Expression levels for proteins co-expressed with ZAPS and ZAPL were quantified using Image J software and normalized to the empty construct control samples. N = 3, * p-value < 0.05 based on Student's t-test.

inhibits progression of virus replication at a stage prior to increased IE2 expression. These results indicate that in the context of viral infection, IE1 expression is unaffected by ZAPS, consistent with suppressed CpG frequency in this transcript enabling evasion of ZAPS.

## Endogenous ZAP is induced during HCMV infection but expression is mutually exclusive to acute virus progression

As overexpression of ZAP profoundly attenuates HCMV replication, we next determined the effects of endogenous ZAP during virus replication. Western blot analysis was performed on total protein lysates harvested from wild type fibroblast cells infected with TB40/E-GFP (MOI of 3), which results in close to 100% infection based on GFP expression. ZAPS expression was robustly induced 24 hours post infection and although levels decreased over time, the levels of endogenous ZAPS remained higher than in uninfected cells throughout the course of the infection (Fig 7A). ZAPL expression was not induced but also decreased over the course of the infection.

To determine whether ZAP expression levels correlated with HCMV infection at the individual cell level, we employed confocal microscopy. Cells were co-stained for ZAP and viral IE1 or IE2 expression in wild type fibroblast cells following infection with AD169, a laboratory adapted strain of HCMV. This strain was used instead of TB40/E-GFP as it does not express GFP, which could interfere with the fluorescent signal. Uniform, low levels of ZAP expression can be seen throughout uninfected cells (Fig 7B). While almost all cells are IE1 positive following infection with HCMV at an MOI of 3, a mixed population of cells demonstrate high and low expression of ZAP. Multiple IE1 positive cells were observed with high levels of ZAP expression, further demonstrating that IE1 expression is not affected by high ZAP levels. In contrast IE2 expression and high levels of ZAP were mutually exclusive, suggesting progression of virus replication is blocked in cells expressing high levels of ZAP or reduction in ZAP is dependent on the progression of viral replication. This is confirmed by quantification of ZAP expression levels in IE1 and IE2 expressing cells (Fig 7C).

The existence of a mixed population of cells expressing high and low levels of ZAP, correlating with progression of virus replication is further supported by sorting of infected cell populations. Following infection with TB40/E-GFP at an MOI of 3, cells were sorted into high and

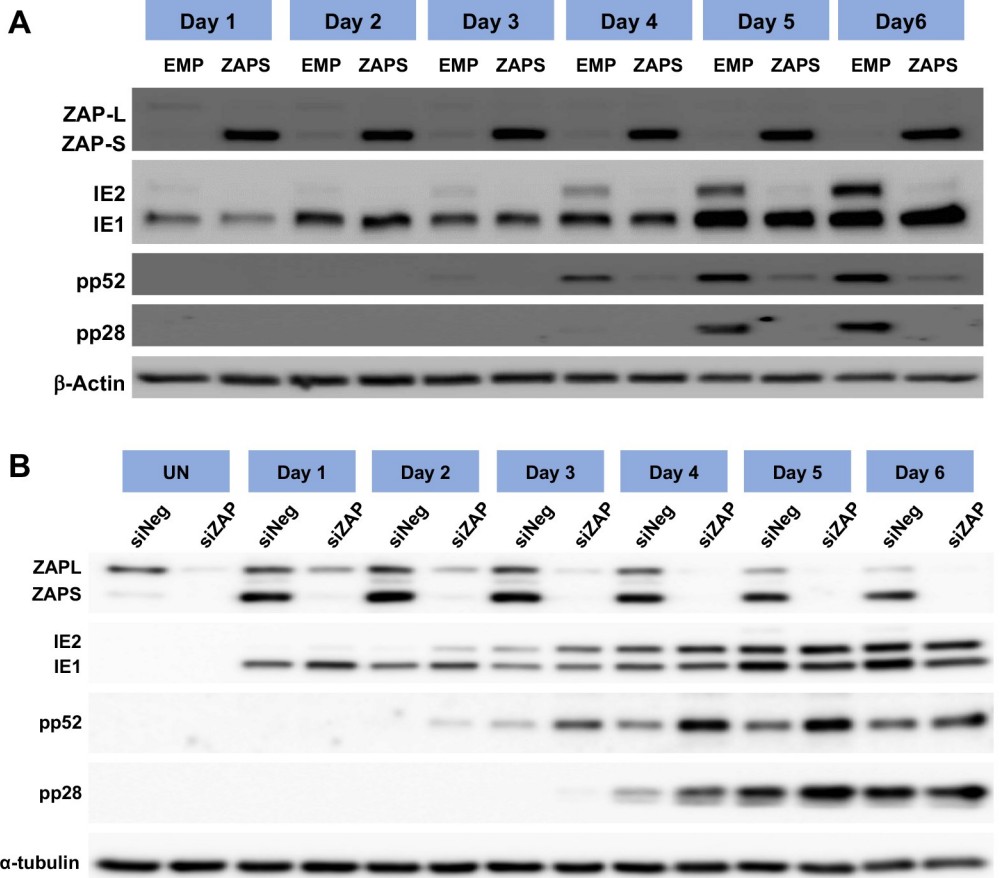

**Fig 6. IE1 expression is not affected by ZAP overexpression.** (A) Wild type (WT) fibroblast cells were transduced with ZAPS or empty vector control lentiviruses (EMP) and infected with TB40E-GFP 48 hours post transduction. Total protein lysates were harvested every 24 hours and the expression levels of the viral proteins of each major kinetic class of HCMV were determined by western blot analysis. The result demonstrates that ZAPS overexpression leads to significant reduction of IE2 expression and downstream early and late viral proteins, whereas IE1 expression remains unaffected. (B) Fibroblast cells were transfected with a negative control siRNA (siNeg) or an siRNA against ZAP (siZAP) and infected two days post transduction. Total protein was harvested at the indicated time points and analysed by western blot analysis for viral protein expression.

low GFP expressing populations, 24 hours post infection, then reseeded and total protein levels determined at the indicated time points (Fig 8). Western blot analysis indicates that the low GFP expressing population correlated with high levels of ZAPS expression and a failure in progression of virus replication (although IE1 protein expression could still be detected). In contrast, high GFP expressing populations corresponded with lower ZAPS expression and high levels of viral protein production, consistent with efficient virus replication.

These data suggest that high levels of ZAP expression are mutually exclusive with the successful progression of virus replication. In contrast, IE1 expression is unaffected, consistent with suppressed CpG content facilitating evasion of ZAP detection.

## TRIM25 is required for efficient ZAPS induction following HCMV infection or IFN treatment

TRIM25 is recognised as a key functional partner of ZAPS antiviral activity. However, the mechanism by which TRIM25 augments ZAPS antiviral activity is unclear. It has been shown

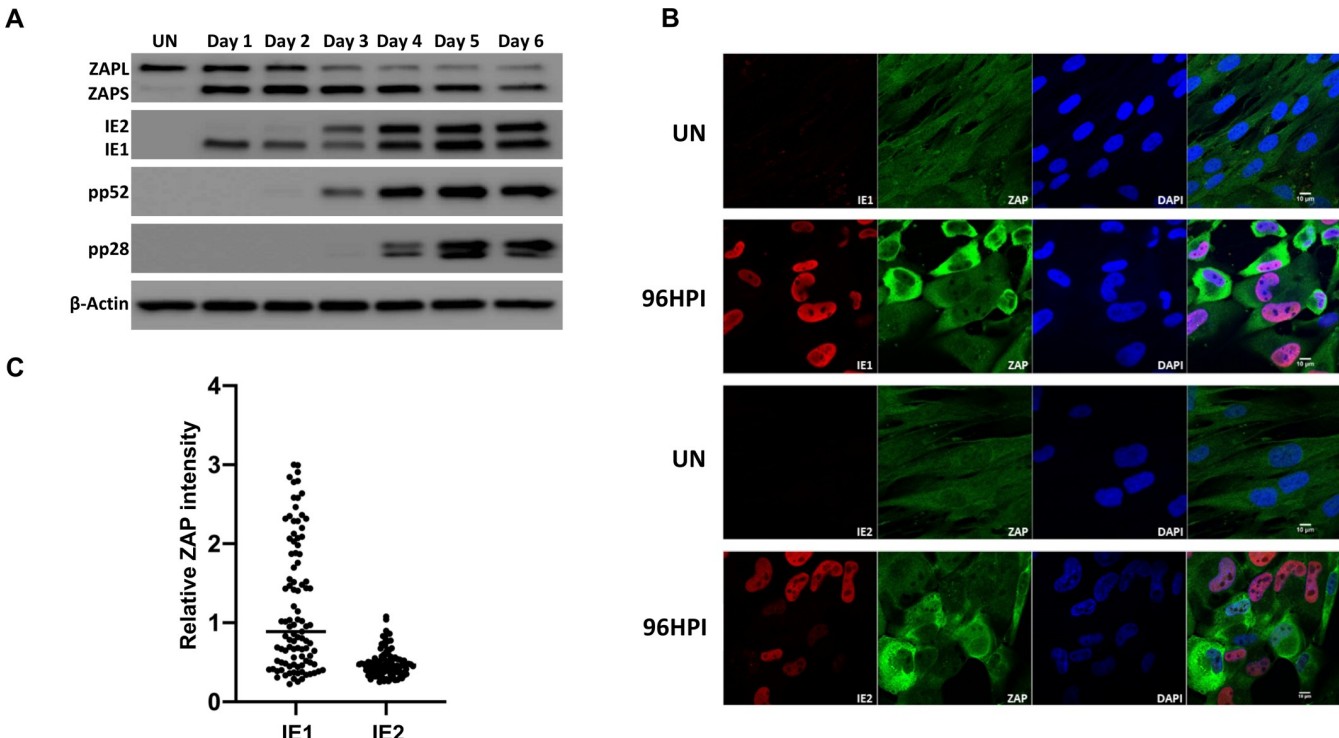

**Fig 7. ZAP expression is reduced in IE2 positive cells.** (A) Wild type fibroblast cells were infected with TB40E-GFP at an MOI of 3. Total protein lysates were harvested every 24 hours and the expression levels of ZAP and the viral proteins of each major kinetic class of HCMV were determined by western blot analysis. It demonstrates that ZAPS expression was robustly induced 24 hours post infection and remained higher than in uninfected cells throughout the course of the infection. (B) Wild type fibroblast cells were infected with HCMV 96 hours post infection, cells were fixed and stained with ZAP along with either IE1 or IE2. ZAP expression levels in IE1 or IE2 expressing cells were determined by confocal microscopy. Nuclei are stained with DAPI. (C) ZAP expression levels in 100 IE1-expressing cells and 100 IE2-expressing cells at 96 HPI were quantified and normalized to ZAP expression levels in uninfected cells from the confocal images using ZEN blue software.

that TRIM25 ubiquitinates ZAPS and itself through its E3 ubiquitin ligase function [18, 19]. However, ZAPS antiviral activity does not seem to be dependent on TRIM25 ubiquitination. In this study, we show that like ZAPS, TRIM25 is a potent, IRF3 independent inhibitor of HCMV replication (Figs 1 and 2).

Strikingly, in addition to increasing late viral protein expression, siRNA knockdown of TRIM25 results in a substantial reduction in ZAPS expression following infection with HCMV and a corresponding increase in ZAPL expression (Fig 9A). As previously described, ZAPS and ZAPL are expressed from the same primary transcript through differential splicing and only ZAPS expression is considered to be stimulated by IFN treatment, indicating that differential splicing is involved in the regulation of ZAPS and ZAPL levels (Fig 9B) [58].

To determine whether the failure in ZAPS induction following TRIM25 knockdown was at the RNA or protein level, ZAPS and ZAPL RNA levels were determined by RT-qPCR. ZAPS and ZAPL have independent 3'UTRs allowing design of primers that can differentiate between the two transcripts. Fibroblast cells were transfected with a control siRNA or siRNA targeting TRIM25 and infected with TB40/E-GFP. Total RNA was harvested at 24 and 48 HPI and primers specific to ZAPS, ZAPL or to a shared region of ZAP were used to measure transcript levels. While total levels of ZAP RNA were unaffected by TRIM25 knockdown, levels of ZAPS were reduced with a corresponding increase in ZAPL levels (Fig 9C, 9D and 9E). This effect is not specific to HCMV infection as the same observation can be seen in uninfected fibroblast cells and cells treated with IFN (Fig 9D and 9E). Furthermore, ZAP induction, as well as induction

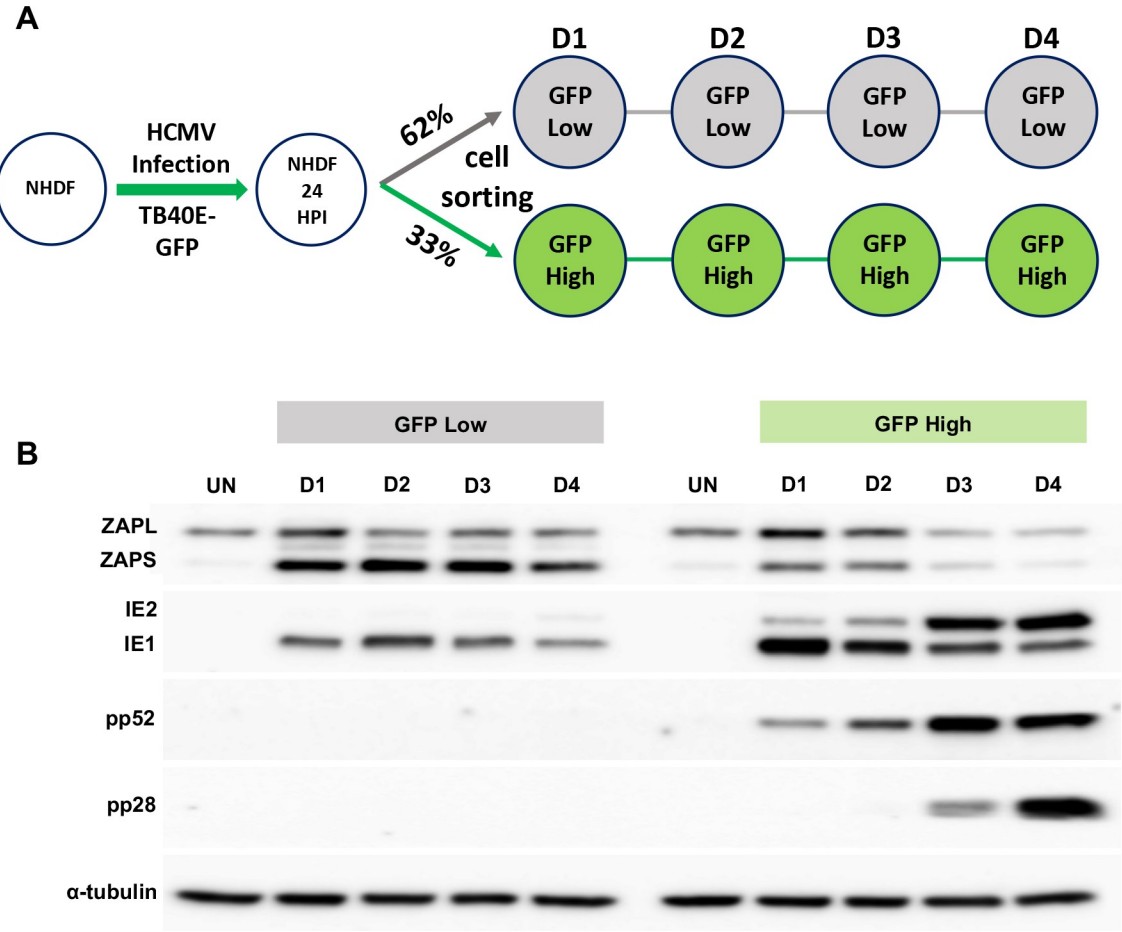

**Fig 8. High ZAP expression correlates with failure in HCMV acute replication.** (A) Diagram of cell sorting following TB40E-GFP infection. 24 hours post infection, fibroblast cells were sorted into low and high GFP expressing populations, and then re-seeded. Total protein lysates were harvested at the indicated time points. (B) Western blot analysis shows that the low GFP expressing population correlated with high levels of ZAPS expression and a failure in progression of virus replication, although IE1 expression can still be detected. In contrast, the high GFP expressing population correlated with low ZAPS expression and high levels of viral protein production, consistent with successful virus replication.

of other well characterised ISGs, was not reduced by TRIM25 knockdown ruling out a general failure in IFN induction (Figs 9 and S8). The results clearly show that efficient IFN induced gene expression occurs despite TRIM25 knockdown, suggesting the effect is specific to differential splicing of ZAP. Although these results do not rule out the possibility that TRIM25 regulates the antiviral function of ZAPS through direct protein-protein interaction, they explain how TRIM25 contributes to the activity of ZAPS by regulation of ZAPS induction through differential splicing.

## Discussion

Associated pathologies and therapeutic potential of HCMV is dependent on the host immune response against the virus. As IFNs shape the innate and adaptive responses to the virus, understanding how the IFN response is regulated during HCMV infection and how the virus subverts this response could have important implications for our understanding of diseases associated with the virus as well as for the rational design of vaccines and cancer therapeutics.

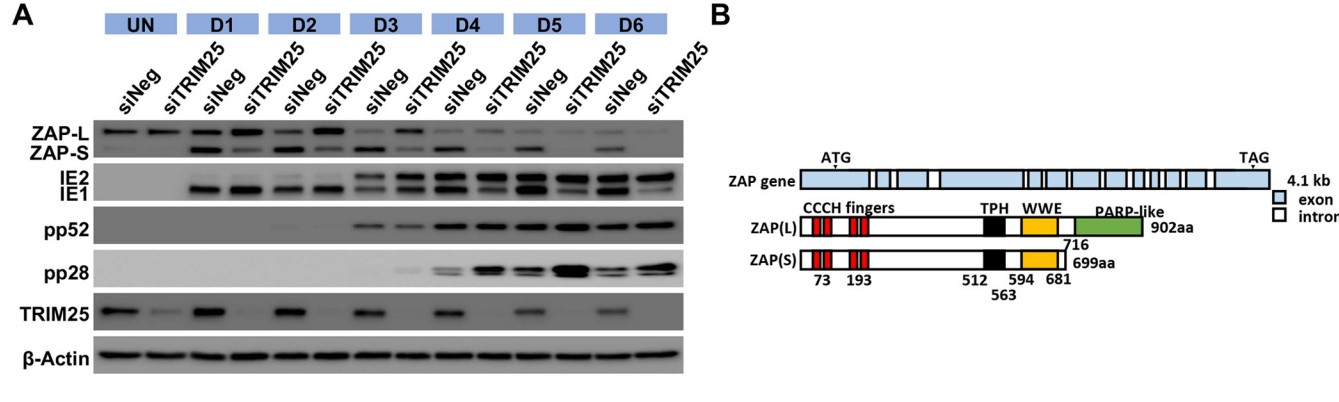

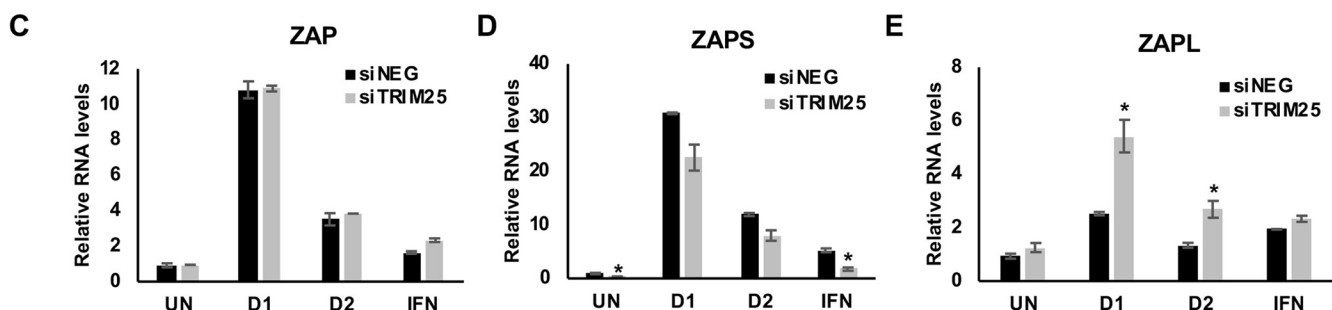

**Fig 9. TRIM25 regulates differential splicing of ZAP.** (A) Wildtype fibroblast cells were transfected with TRIM25 siRNA or negative control siRNA (siNEG) and infected with TB40/E-GFP 48 hours post transfection (MOI of 3). Total protein lysates were harvested every 24 hours and expression levels of ZAPS and ZAPL and viral proteins were determined by western blot analysis. The result demonstrates that knockdown of TRIM25 leads to substantial reductions in ZAPS expression and corresponding increase in ZAPL expression. (B) Diagram of the genomic structure of human ZAP gene showing the two major isoforms, ZAPS and ZAPL. Wildtype fibroblast cells were transfected with TRIM25 siRNA or negative control siRNA (siNEG) and infected with TB40E-GFP at an MOI of 3, uninfected or treated with IFN-a. Total RNA was harvested from infected samples one (D1) and two (D2) days post infection with mock and IFN treated cells harvested one day post treatment. Primers to a ZAP shared exon (C) or specific primers to ZAPS (D) and ZAPL (E) were used to determined transcript levels by quantitative RT-qPCR analysis. Levels were normalised to GAPDH and compared to RNA levels from cells transfected with the control siRNA. N = 2, * p-value < 0.05 based on Student's t-test.

Although it is clear that the IFN response is important during HCMV infection, which of the hundreds of induced ISGs play a critical role is less clear. Here we present a systematic analysis, determining the effect of over 400 individual ISGs on HCMV replication. By combining screens in wild type and IRF3 knockout cells we were able to define specific subsets of ISGs that were IRF3 independent and therefore more likely to represent specific inhibitors of virus replication. These included IDO and RIPK2, which have previously been identified as important antiviral factors during HCMV infection [47, 48]. In addition, multiple novel inhibitors were identified, including ZAP and TRIM25, which have previously been shown to act in a coordinated antiviral fashion [18, 19].

ZAP was originally identified as an antiviral protein by screening a rat cDNA library for factors that inhibited the replication of Moloney murine leukemia retrovirus [13]. Subsequent studies have demonstrated that ZAP has antiviral activity against a range of RNA viruses, including HIV, filoviruses, flaviviruses, coxsackievirus B3, influenza A virus, Newcastle disease virus and Hepatitis B virus, a partially double stranded DNA virus [50–53, 59–61]. Initial studies indicated that ZAP antiviral activity was based on direct binding to viral RNA and degradation through recruitment of exosome complex components and inhibition of translation [13, 14, 56]. Our studies show that ZAPS and ZAPL can inhibit expression of HCMV genes with

high CpG ratios and rather than being a consequence of RNA degradation, this inhibition appears to be post transcriptional, as RNA levels of target transcripts were not reduced (Fig 4F). Further studies have suggested that ZAPS interacts with RIG-I to stimulate IFN expression through IRF3 signalling [60]. This is consistent with the partial rescue of ZAPS inhibition of HCMV in IRF3 KO cells observed in this study (Fig 2B), suggesting a dual functionality of ZAPS in HCMV antiviral activity, acting both as a pathogen sensing protein and as a specific antiviral factor.

Until recently, the sequence or sequence characteristics that defined ZAPS RNA binding specificity were not known. However, a study using a focused siRNA screen identified ZAPS and TRIM25 as host factors responsible for inhibition of HIV-1 constructs with artificially raised CpG dinucleotide frequencies and it was shown that ZAPS specifically binds to RNA sequences at CpG dinucleotide motifs [13]. The molecular mechanism for this binding has now been established through X-ray crystallography of the ZAP RNA binding domain complexed with a target CpG RNA, which demonstrates the second of four zinc fingers creates a highly basic patch that is required for specific binding of CpG dinucleotides [15, 62].

It had previously been suggested that mammalian RNA and small DNA viruses suppress CpG dinucleotides to mimic the composition of their host genomic makeup, thereby avoiding recognition as foreign nucleic acid [6–8]. While CpG bias has previously been reported for herpesviruses, this bias was attributed to potential methylation status rather than evasion of a host antiviral response [54]. The pattern of CpG dinucleotide frequencies in herpesviruses is so distinctive that, with few exceptions, viruses can be attributed to the main three subfamilies based on the CpG dinucleotide frequency patterns within open reading frame (S1–3 Figs). The majority of alpha-herpesviruses demonstrate little or no CpG suppression, while gamma-herpesviruses demonstrate substantial suppression across the genome. Beta-herpesviruses display the most striking pattern of all, with suppression of CpG dinucleotides linked to temporal gene expression and specifically restricted to IE1. Scanning analysis shows that the IE1 coding region is the only region of the HCMV genome that is suppressed for CpG content [54]. It is intriguing to speculate that the genomes of these large DNA viruses may have been so dramatically moulded by a single host antiviral mechanism, although other factors may also play a role.

The fact that CpG dinucleotide patterns are so uniform across the virus sub-families also suggests a central underlying biological relevance. For example, alpha-herpesviruses are mainly associated with latent infections in neuronal cells where the interferon response may be limited due to immune privilege and constitutive expression levels of ZAP are low [20], whereas beta-herpesviruses and gamma-herpesviruses are mainly associated with latency in haematopoietic cells and are therefore under greater pressure to evade host cell IFN responses [63]. Alternatively, alpha-herpesviruses may have evolved a robust mechanism that directly counteracts ZAP antiviral activity, making it unnecessary to repress CpG dinucleotides. This is supported by a previous study that showed HSV-1 was able to replicate to wild type levels despite expression of a ZAP construct, although the construct expressed a truncated version of rat ZAP, rather than full length human ZAP [64]. Further studies will be required to determine whether the CpG dinucleotide frequency in herpesvirus sub-families is directly related to ZAP expression and why these patterns are so distinctively associated with the specific virus sub-families. Whether herpesviruses express countermeasures against ZAP and TRIM25 will also be an important area of investigation.

Multiple studies have linked TRIM25 with efficient ZAP antiviral activity, and, like ZAPS, our screen identified TRIM25 as an IRF3 independent inhibitor of HCMV. TRIM25 is a ubiquitin E3 ligase that catalyses ubiquitylation and ISGylation of target proteins [16, 17]. TRIM25 contains a zinc ring finger, B-box, coiled coil and PRY/SPRY domain. Both Takata *et al* and Li *et al* demonstrated that TRIM25 is required for ZAP antiviral activity through

siRNA screens [11, 18], whereas Zheng *et al* identified TRIM25 as a ZAP interacting factor through affinity purification [19]. These studies demonstrated that TRIM25 interacts directly with ZAP through the PRY/SPRY domain and ubiquitinates ZAP and itself. While ubiquitination requires binding of ZAP and TRIM25 to RNA it does not seem to be directly necessary for ZAP antiviral activity [18, 19, 65]. TRIM25 also inhibits influenza A virus in a ZAP independent manner through direct binding to the viral ribonucleoprotein complex [66]. We show that TRIM25 expression potently inhibits HCMV replication although further studies will be required to determine whether this occurs in a ZAP dependent or independent manner. In addition to TRIM25, Ficarelli *et al.*, identified the cellular factor KHNYN as an important cofactor of ZAP [67], suggesting multiple host factors are likely to be involved in recognition of CpG motifs and further analysis will be required to fully characterise how each host factor contributes to this antiviral mechanism.

In addition, our studies suggest that TRIM25 regulates the alternative splicing of ZAP. A recent study demonstrated that alternative splicing of the ZAP primary transcript leads to at least four different isoforms; ZAPS, ZAPM, ZAPL and ZAPXL [58], although our study focused on the major isoforms ZAPS and ZAPL. ZAPL contains a catalytically inactive PARP-like domain at the C terminus, that is missing from ZAPS. Both ZAPS and ZAPL have been reported to have antiviral activity, although ZAPS was identified as the host factor responsible for CpG recognition [13]. In our study we found that both ZAPS and ZAPL could target high CpG transcripts. Following virus infection or IFN treatment ZAP is induced, however differential splicing results in higher levels of induction of ZAPS compared to ZAPL. Following HCMV infection, TRIM25 knockdown resulted in significantly lower levels of ZAPS protein and RNA levels with a corresponding increase in ZAPL levels indicating a TRIM25 dependant change in differential splicing of ZAP. Regulation of ZAP splicing by TRIM25 occurs in uninfected cells and IFN stimulated cells, indicating it is not specific to HCMV but rather a fundamental aspect of ZAP regulation. It is unclear at this point whether TRIM25 plays a direct role in splicing of ZAP or whether the regulation occurs through an intermediate signalling pathway. TRIM25 has been shown to directly bind RNA through its PRY/SPRY domain [65], and plays a role in the maturation of the microRNA let-7 [68], suggesting a more direct role in processing ZAP RNA is possible. However further studies will be required to dissect its precise role [65]. As knockdown of TRIM25 results in upregulation of ZAPL and has been reported to have antiviral activity independent of ZAP, we predict divergent roles of ZAP and TRIM25 in the context of HCMV will be likely.

Increasing the CpG levels within the IE1 coding region could be a viable approach for generating safer live attenuated vaccines with reduced risk of reversion as the mutations could be spread across the entire length of the IE1 gene. Increasing the CpG levels in IE1 could also increase the immunogenicity of the virus by triggering more robust IFN responses, which could improve HCMV vaccines as well as the use of HCMV as a vaccine vector and as a therapeutic cancer vaccine. However, construction of such viruses may be challenging due to the central role the major immediate early genes play in acute replication of the virus and the potential for introduced mutations that disrupt the complex splicing events or regulatory regions embedded within the coding region of IE1. Future experiments will be required to determine the potential application of such viruses.

## Materials and methods

### Cell culture and viral infection

Normal Human Dermal Fibroblasts (NHDF; Gibco) and IRF3 -/- cells were maintained in Dulbecco's modified high glucose media (DMEM; Sigma) supplemented with 10% fetal bovine

serum (FBS; Gibco) and 1% penicillin-streptomycin (Invitrogen). IRF3 -/- cells were provided by Victor DeFillipis. A low passage HCMV strain TB40E-GFP [45], which expresses GFP from an SV40 promoter was used for arrayed ISG expression library screening, western blot analysis, Real-Time PCR, and flow cytometry analysis. Laboratory adapted HCMV strain AD169 was used for immunofluorescence experiments.

## Arrayed ISG lentivirus expression library screening

The human lentivirus expression library has been previously described [44]. In brief the library encodes 420 human ISG genes on the pSCRPSY backbone (KT368137.1). Normal human dermal fibroblast cells and IRF3-/- cells were seeded in 96-well plates a day before transduction. Next day, cells reached 90–95% confluency and were transduced with the ISG library. Transduced cells were incubated for 48 hours and then infected with GFP expressing TB40E virus at an MOI of three. GFP intensity was monitored every 24 hours with Synergy HT microplate reader (Biotek).

## siRNA transfection

NHDFs were seeded in 6-well plates a day before siRNA transfection. Next day, cells reached 90–95% confluency and were transfected with siRNA twice (4 hours apart between first and second transfections) using Lipofectamine RNAiMAX (Invitrogen) according to the manufacturer's protocol. siRNAs used in this article are Human ON-TARGETplus siRNAs against TRIM25 and ZC3HAV1 (4 individual siRNAs per gene; Dharmacon). Transfected NHDFs were incubated for 48 hours and then infected with GFP expressing TB40E virus at an MOI of three.

## Western blot analysis

Cells were lysed with RIPA buffer (0.1% SDS, 1% Triton X-100, 1% deoxycholate, 5 mM EDTA, 150 mM NaCl, and 10 mM Tris at pH 7.2) containing protease inhibitor cocktail (Roche). Ten micrograms of the total lysate was separated in 10% SDS-polyacrylamide gels and transferred to PVDF membranes (Millipore). Primary antibodies used in this paper are mouse anti-CMV IE1/2 monoclonal antibody (MAB8131, Millipore), mouse anti-CMV pp52 monoclonal antibody (CH16, Santa Cruz Biotechnology), mouse anti-CMV pp28 monoclonal antibody (CH19, Santa Cruz Biotechnology), rabbit anti-ZAP polyclonal antibody (PA5-31650, Invitrogen), mouse anti-TRIM25 monoclonal antibody (BD Biosciences), rabbit anti-T7 tag monoclonal antibody (D9E1X, Cell Signaling Technology), mouse anti-alpha tubulin monoclonal antibody (DM1A, Abcam) and mouse anti-β-Actin monoclonal antibody (Abcam). Blots were probed with primary antibody (1:500–1:5000) diluted in 5% dehydrated milk in Tris Buffered Saline (TBS) and subsequently the HRP-conjugated secondary antibodies (Pierce) at 1:5000. Blots were washed in TBS three times, incubated with chemiluminescent substrate (SuperSignal West Pico; Thermo Scientific) according to the manufacturer's protocol, and exposed in G:Box (Syngene) for visualization of bands.

## Vector Construction and transfection

The expression vectors used in the article were made by cloning the coding sequences of UL123 (IE1, TB40E), UL122 (IE2, TB40E), UL44 (pp52, TB40E), Human GAPDH with double T7-tag at the N-terminus, into the pcDNA3.1 vector. The ZAP(L) and ZAP(S) expression vectors were made by cloning the coding sequences of ZAP(L) and ZAP(S) into the pcDEF3.1 vector derived from pcDNA3.1 with the CMV promoter replaced by the EF1alpha promoter.

## Immunofluorescence

Laboratory adapted HCMV strain AD169 infected cells were fixed in 4% paraformaldehyde solution for 20 minutes and then permeabilized in Methanol:Acetone solution (1:1) at -20°C for 7 minutes, and then blocked with 5% human serum in PBS for 30 minutes. Primary and secondary antibodies were diluted with 5% human serum in PBS. Cells were washed with PBS after primary and after secondary antibody incubations. Primary antibodies used in this paper are mouse anti-CMV IE2 monoclonal antibody (12E2, Santa Cruz Biotechnology), mouse anti-CMV IE1/2 monoclonal antibody (MAB8131, Millipore), and rabbit anti-ZAP polyclonal antibody (PA5-31650, Invitrogen) at 1:500. Alexa-fluor-647 conjugated goat anti-mouse or Alexa-fluor-488 conjugated goat-anti-rabbit IgG secondary antibodies were diluted 1:1000. All images were acquired with Zeiss LSM 710 confocal microscope fitted with 63X/1.4 oil-immersion objective lens.

## Real-Time PCR analysis

Total RNA was isolated by using Trizol solution according to the manufacturer's protocol followed by DNase (Turbo DNA-free kit, Ambion) treatment, and then reverse transcribed with poly T primers using High Capacity cDNA Reverse Transcription Kit (Invitrogen). Real-Time PCR was carried out using by Taqman assays with pre-designed gene-specific primer/probe set (Applied Biosystems) on Rotor gene 3000 (Corbet Research). Custom primer/probe set are TCCTCTCTCAGGATCTGTATGT, GGAGAGGAAGGAGTCAAAGATG, and 56-FAM/ACCATCTAC/ZEN/CCATTGGCTCAAGCA/3IABkFQ for ZAP(L), AGCATGGTGTGACT GAA AGG, CTTCACAGCTGGAGAAGCTAAA, and 56-FAM/TCTGAAAGG/ZEN/GAA GTCTGAG CGAGTCT/3IABkFQ for ZAP(S), CGTCAAACAGATTAAGGTTCGAGTGG, CCACATC TCCC GCTTATCCTCG, and 56-FAM/CATGCTCTG/ZEN/CATAGTTAGCC CAATACACTTCATCT CCTCG/3IABkFQ for IE1, ATGGTGCGCATCTTCTCCACC, TTACTGAGACTTGTTCCTCA GGTCCTG, and 56-FAM/CAGGCTCAG/ZEN/GGTGT CCAGGTCTTCGGGAGG/3IABkFQ for IE2, and CAAGGACCTGACCAAGTTCTAC, GCCGAGCTGAACTCCATATT and 56-FAM/CATGGAGAT/ZEN/CTTGGCCGACA GGTC/3IABkFQ for pp52.

## Fluorescence-activated cell sorting (FACS)

NHDFs were infected with GFP expressing TB40E virus at an MOI of three. 24 hours later, the cells were trypsinized, and resuspended in PBS buffer. The resuspended cells were then sorted into low GFP (parameter set using uninfected cells) or high GFP population by the BD FACSAria IIIu cell sorter. Immediately after sorting, a small portion of the cells from each population were collected and lysed with RIPA buffer, and the remaining low GFP and high GFP cells were re-seeded into a 6-well plate respectively and incubated for another 24 to 72 hours before lysed with RIPA buffer.

## Analysis of CpG dinucleotides frequency in Herpesvirus genome

CpG dinucleotide frequencies were determined using the program Composition scan in the SSE Sequence editor v1.3 (PMID: 22264264). Results were expressed as ratios of observed frequencies (fCpG) to those predicted by the frequencies of their component mononucleotides (fC x fG). This ratio was normalised through computing constraints imposed by amino acid coding on dinucleotide frequencies. For example, a methionine codon enforces the presence of ApU and UpG dinucleotides, while glycine codons requires a GpG dinucleotide, as well as GpN, Corrected dinucleotide ratios are therefore based on observed to expected frequencies coding-enforced dinucleotides are excluded. All accession data is listed in S2 Table.

## Supporting information

**S1 Fig.** (A) Direct comparison of relative HCMV primary replication for each individual ISG between wild type (WT) and IRF3 knockout (KO) fibroblast cells. Blue box = ISGs that reduced virus production by more than 2-fold in wild type cells, but were not in IRF3 KO cells. The green box = ISGs that reduced virus production by more than 2-fold in both wild type and IRF3 KO cells. Relative primary replication and virus production levels of HCMV for the ISGs in the blue box (B) and for ISGs that reduced primary replication by more than 50% (C) were plotted.
(TIF)

**S2 Fig.** ZAPS and TRIM25 overexpression inhibits HCMV replication (A) Fibroblast cells were transduced with a two-fold dilution series of control empty lentivirus (control) or lentivirus expressing ZAPS or TRIM25. Cells were infected with TB40E-GFP at an MOI of 3 with total protein samples harvested 24 HPI and subjected to western blot analysis for ZAP and TRIM25 levels. (B) GFP levels were measured by plate fluorometry seven days post infection for the corresponding dilution series. GFP levels are shown as a percentage of infected cells transduced with a corresponding serial dilution of a control empty lentivirus (N = 3). Human Epithelial cells (RPE-1) (C) or human umbilical vein endothelial cells (HUVEC) (D) were transduced with a control empty lentivirus (EMP) or lentivirus expressing ZAPS and TRIM25, then infected with TB40E-GFP at an MOI of 3 (titred on corresponding cell line). N = 3.
(TIF)

**S3 Fig. Efficient knockdown of ZAP and TRIM25 by siRNA.** Fibroblast cells were transfected with siRNA targeting ZAP (siZAP), TRIM25 (siTRIM25) or a negative control siRNA (siNeg). Total protein was harvested three days post transfection and levels of ZAP (A) and TRIM25 (B) determined by Western blot analysis. Pools of four siRNAs against ZAP (C) and TRIM25 (D) were deconvoluted and tested individually for effect on HCMV replication. Levels of virus in supernatant were tested seven days post infection with GFP levels normalised to cells transfected with a negative control siRNA. N = 4, * p-value < 0.05 based on Student's t-test.
(TIF)

**S4 Fig.** CpG (A) and GpC (B) corrected ratios for ORFs of alpha-herpesviruses. Corrected CpG and GpC ratios were calculated for alpha-herpesviruses as described in the materials and methods and in Fig 3.
(TIF)

**S5 Fig.** CpG (A) and GpC (B) corrected ratios for ORFs of gamma-herpesviruses. Corrected CpG and GpC ratios were calculated for gamma-herpesviruses as described in the materials and methods and in Fig 3.
(TIF)

**S6 Fig.** CpG (A) and GpC (B) corrected ratios for ORFs of beta-herpesviruses. Corrected CpG and GpC ratios were calculated for beta-herpesviruses as described in the materials and methods and in Fig 3.
(TIF)

**S7 Fig. Alignment of WT IE1 exon 4 and high CpH IE1 exon 4.** Exon 4 of IE1 was synthesised with synonymous mutations introduced that create CpG motifs. Synonymous mutations indicated below the WT sequence.
(TIF)

**S8 Fig. TRIM25 knockdown does not inhibit ISG induction.** Wild-type fibroblast cells were transfected with TRIM25 siRNA or negative control siRNA (siNEG). 48 hours later, the cells were infected with TB40E-GFP at an MOI of 3. Total RNA was harvested at 24 hours post infection and the RNA levels of (A) ISG20, (B) MDA5 and (C) RIG-I were determined by RT-qPCR analysis and compared to uninfected (UN) cells harvested at the same time point. The result demonstrates that ISG induction following HCMV infection is not inhibited by TRIM25 knockdown. N = 2.
(TIF)

**S1 Table. Percentage inhibition of virus replication for ISG screens.** Primary fibroblast cells (WT) or IRF3 KO fibroblast cells were transduced with a human arrayed ISG lentivirus expression library and infected with TB40E-GFP at an MOI of three, two days post transduction. GFP expression levels were measured seven days post infection (D7) by plate fluorometry and supernatant transferred to untransduced primary fibroblast cells to measure virus production (R4). GFP expression determined as a percentage of the plate average. Two independent screens were performed (A and B).
(XLSX)

**S2 Table. CpG ratios for Herpesviruses.** CpG dinucleotide frequencies were determined using the program Composition scan in the SSE Sequence editor v1.3 (PMID: 22264264). Results were expressed as ratios of observed frequencies (fCpG) to those predicted by the frequencies of their component mononucleotides (fC x fG). This ratio was normalised through computing constraints imposed by amino acid coding on dinucleotide frequencies.
(XLSX)

## Acknowledgments

We would like to acknowledge help and technical assistance from Robert Fleming and Graeme Robertson of the Roslin Bioimaging and flow cytometry facility and Helen Brown and James Prendergast for critical review of the manuscript.

## Author Contributions

**Conceptualization:** Peter Simmonds, Sam J. Wilson, Finn Grey.

**Data curation:** Yao-Tang Lin, Stephen Chiweshe, Dominique McCormick.

**Formal analysis:** Yao-Tang Lin, Stephen Chiweshe, Dominique McCormick, Anna Raper.

**Funding acquisition:** Peter Simmonds, Sam J. Wilson, Finn Grey.

**Investigation:** Yao-Tang Lin, Stephen Chiweshe, Dominique McCormick, Eleanor Gaunt.

**Methodology:** Anna Raper, Arthur Wickenhagen, Victor DeFillipis.

**Resources:** Peter Simmonds, Sam J. Wilson, Finn Grey.

**Software:** Peter Simmonds.

**Supervision:** Finn Grey.

**Validation:** Yao-Tang Lin.

**Writing – original draft:** Yao-Tang Lin, Finn Grey.

**Writing – review & editing:** Eleanor Gaunt, Peter Simmonds, Sam J. Wilson.

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
