## [Decision Letter · Decision Letter 0]

28 Feb 2020

Dear Career Track Fellow Grey,

Thank you very much for submitting your manuscript "Human cytomegalovirus evades ZAP detection by suppressing CpG dinucleotides in the major immediate early genes" for consideration at PLOS Pathogens. As with all papers reviewed by the journal, your manuscript was reviewed by members of the editorial board and by several independent reviewers. In light of the reviews (below this email), we would like to invite the resubmission of a significantly-revised version that takes into account the reviewers' comments.

Dear Dr Grey,

Finn, I hope this finds you well. We have received reviews back from three highly competent herpesvirologists who provided feedback on your manuscript. It should be noted that all three (and myself) were impressed with the original ISGs screen that you performed. As you can see, the opinions of the reviewers differed and all three had significant concerns that they would like addressed prior to the designation of this as an acceptable manuscript. If you believe you can address the concerns in a timely fashion, I would be more than happy to revisit your modified manuscript for consideration.

Yours,

Eain

Eain A. Murphy Ph.D.

We cannot make any decision about publication until we have seen the revised manuscript and your response to the reviewers' comments. Your revised manuscript is also likely to be sent to reviewers for further evaluation.

Sincerely,

Eain A Murphy, Ph.D.

Associate Editor

PLOS Pathogens

Blossom Damania

Section Editor

PLOS Pathogens

Kasturi Haldar

Editor-in-Chief

PLOS Pathogens

orcid.org/0000-0001-5065-158X

Michael Malim

Editor-in-Chief

PLOS Pathogens

orcid.org/0000-0002-7699-2064

Dear Dr Grey,

Finn, I hope this finds you well. We have received reviews back from three highly competent herpesvirologists who provided feedback on your manuscript. It should be noted that all three (and myself) were impressed with the original ISGs screen that you performed. As you can see, the opinions of the reviewers differed and all three had significant concerns that they would like addressed prior to the designation of this as an acceptable manuscript. If you believe you can address the concerns in a timely fashion, I would be more than happy to revisit your modified manuscript for consideration.

Yours,

Eain

Eain A. Murphy Ph.D.

Reviewer's Responses to Questions

**Part I - Summary**

Reviewer #1: Lin, et al. from the group of Finn Grey employ an interferon stimulated gene (ISG) screen in normal human dermal fibroblasts to identify host restriction factors against human cytomegalovirus (HCMV). They uncover the interferon induced ZAPS and TRIM25 as factors that reduce HCMV replication, which is suggested to occur in an IRF3-independent fashion. Upon a series of overexpression and siRNA-mediated knockdown experiments, the authors suggest that ZAPS and TRIM25 restrict HCMV replication and that this occurs via ZAPS detection of CpG dinucleotides in HCMV transcripts. They further suggest that the major immediate early HCMV transcript IE1 has evolved to evade ZAP detection via an underrepresentation of CpG dinucleotides. They show that TRIM25 depletion impacts the alternative splicing of ZAP to generate ZAPS, implicating TRIM25 restriction of HCMV through ZAPS activity. While the study provides an interesting link between ZAP and TRIM25-mediated CpG dinucleotide recognition of a large DNA virus, the experimental rigor required to draw this conclusion is lacking. The manuscript could be greatly improved with the addition of adequate biological replicates, including for the ISG screen and quantified immunoblots, and a more detailed assessment of the mechanism of ZAPS in restricting HCMV replication. Unfortunately, the manuscript is further limited by incomplete interpretation of the results and exclusive use of transient siRNA as a protein depletion strategy. The results show unreliable reproducibility of phenotypes that do not always align with the proposed model. In certain cases, equally viable alternative hypotheses were not discussed, suggesting that they were not considered. At times the effects of ZAPS and TRIM25 on HCMV replication were over-stated and not sufficiently supported by rigorous assessments. The manuscript also lacks mechanistic insights into the mechanism of restriction.

Reviewer #2: In this manuscript, Li et al present intriguing evidence for a novel mechanism of DNA virus detection by the intrinsic immune system. They screened for interferon (IFN)-stimulated genes (ISGs) to identify ZAP and TRIM25 as direct inhibitors of human cytomegalovirus (hCMV) gene expression and replication. ZAP and TRIM25 were recently discovered as restriction factors for RNA viruses with high CpG dinucleotide content, but there is very little information on the relevance of these proteins for DNA viruses. They go on to show that the hCMV genome has evolved to suppress CpG dinucleotides selectively in the major immediate-early 1 (IE1) encoding reading frame. They provide strong evidence that the IE1-specific suppression serves to evade detection by ZAP at very early times of infection (when ZAP expression is induced), while ZAP and TRIM25 are down-regulated at later times. They further show that TRIM25 regulates differential expression of ZAP isoforms during hCMV infection and following IFN treatment of non-infected cells.

This is a well-designed, very timely and overall convincing study that provides a plausible explanation for the long-standing observation of selective CpG suppression at the hCMV major IE locus.

Reviewer #3: This manuscript by Lin and colleagues reports that the host ISGs ZAP and TRIM25 can both function to directly inhibit replication of human cytomegalovirus in infected fibroblasts. These two ISGs were identified as direct inhibitors by performing an arrayed screen overexpressing all known ISGs (>400) in both WT and IRF3ko fibroblasts, a clever approach to filter out those ISGs which could inhibit in this experimental system by simply inducing IFN-I expression through their own overexpression. Although ZAP has been shown in the past to restrict expression/replication of RNA viruses through interaction with CpG dinucleotide rich regions within their genomes, a role for this PRR in the inhibition of large DNA viruses has not been previously demonstrated. It has been known for several decades that the alpha, beta and gamma herpesviruses show significant differences in the levels of CpG dinucleotide repeats within their respective genomes. This suggests that there are unique selective pressures driving this genomic variation in the different viral subfamilies, but to this point the potential reasons for this have been purely speculative. The results reported here provide the first evidence that these differences in CpG levels very likely impact the sensitivity of different herpesviruses to regulation via ZAP-dependent inhibition, and is therefore a very significant observation.

**Part II – Major Issues: Key Experiments Required for Acceptance**

Reviewer #1: 1) Abstract, line 28 and Results, line 127: Is TRIM25 being described as a direct inhibitor here? Evidence is lacking. Lines 140-144: The description of “direct inhibitor” is unclear in these contexts of IFN-independent signaling (really just IRF3-independent), or physically direct interactions with components of the virus.

2) Figure 1/Supplemental Figure 1/Supplemental table 1: Figure 1 identifies a set of ISGs that reduce GFP expression (HCMV production) in both WT cells and IRF3KO cells. This suggests that several of the ISGs restrict HCMV in an IRF3-independent manner. While not explicitly stated, the ISG screen appears to have been performed as a single biological replicate. If the screen results are expected to serve as a resource to inspire future studies on HCMV pathogenesis and IFN functions, it is important that the authors perform at least one additional biological replicate. This will allow more confident identification of proteins that meet their criteria for ≥2-fold reductions in GFP expression (HCMV production) relative to the average GFP signal in both WT and IRF3 KO cells. Calculations of Z scores, standard deviation ranges, or p values (if biological replicates are performed) may be more informative than just fold-change assessment alone because these would better account for GFP signal variability. In line 149-150 and Suppl. Figure 1, ZAP expression did not inhibit primary replication in the screen (albeit with an n of 1?). Can the authors explain why? The authors should also include a legend for each supplemental table.

3) Figure 2 (and throughout): Figure 2 characterizes the impact of ZAP and TRIM25 on HCMV progeny production, demonstrating that they are restriction factors. Is there any comparison between endogenous and ectopic levels of ZAP expression in overexpression experiments? How do the ectopic levels relate to the levels during infection? Without this comparison it is very difficult to interpret how meaningful the overexpression phenotypes are.

In Figure 2A-B, how do the authors explain the differences in relative GFP fluorescence when ZAPS is overexpressed in WT versus IRF3KO cells? Assuming statistical comparisons relative to the empty vector control, why did cGAS expression statistically significantly reduce GFP fluorescence relative to the empty vector control in IRF3KO cells?

In Figure 2D-E, the data presented are not convincing enough to support the claim that endogenous ZAP has a meaningful impact on the virus. There is very little difference between depletion and the controls. Some differences appear to be well within error. An n of 2 is not sufficient here, should be repeated at least 3 times (full biological repeats). Differences are claimed despite the fact that error bars often overlap. ANOVA is not an appropriate statistical test here. Even a very crude analysis of target versus control at a single relevant time-point would be more meaningful. The authors state that TRIM25 is required for ZAPS antiviral activity. They also state that siRNA depletion of TRIM25 reduces ZAPS expression during HCMV infection (Lines 292-293, Figure 8A) and that this occurs via differential ZAP splicing. Given these findings, one would expect that the effect of ZAPS depletion on HCMV progeny production would be phenocopied by TRIM25 depletion at the same times post-infection. Yet, HCMV progeny production at 72, 96, and 120 hpi (when ZAPS deletion appeared most effective in restricting HCMV) showed little to no difference in siTRIM25 cells compared to siNeg cells. The authors should explain this discrepancy.

4) Figures 4 and 5: Figures 4 and 5 identify a correlation between low ZAP and high IE2 levels in infected cells and demonstrate that IE1 levels are unaffected by ZAP expression. Line 222-224: If ZAPL expression shows the same inhibitory effects as ZAPS expression, then why do the authors believe that upregulation of the short form specifically is so important for restriction of the virus? Immunoblot levels of both ZAP isoforms appear to show that ZAPL is more abundant than ZAPS, further suggesting that both isoforms are capable of antagonizing IE2, pp52, and pp28 protein expression. This should be tested for ZAPL during HCMV infection as in Figure 5.

Was there any attempt to reproduce these phenotypes? Quantification seems of little meaning without reproduction and error bars. These transfection data should also be done with increasing amounts of plasmid to strengthen the anticorrelation. Is this restriction at the level of transcription or translation? Have they looked at mRNA levels? Figure 5 is similar in message to Figure 4 but includes HCMV infection instead of HCMV gene transfection and could really just be part of Figure 4.

5) Figures 6 & 7, and the corresponding results sections: This section of the results aims to establish the effect of endogenous ZAP on viral replication. Despite the fact that ZAP can be knocked down by siRNA (Supplemental Figure 2), the ability of the virus to replicate in ZAP knockdown cells is not investigated further beyond the analysis of viral progeny production presented in Figure 2. Further experimentation to investigate the effect of ZAP knockdown on viral proteins and/or viral DNA replication would do a lot to substantiate the key claims. In addition, an alternate knockdown/knockout approach is crucial to account for the risks of off-target effects, artifacts, and transiency of siRNA-mediated knockdown over six-day long experiments. Unfortunately, the experiments reported in Figures 6 & 7 do very little to mitigate this.

When considering the anti-correlation between ZAPS levels and viral protein levels, it is difficult to interpret what is the cause and what is the effect. The progress of virus infection is often heterologous at a cell to cell level. It is possible that the anti-correlation reflects more successful infection when ZAPS levels are low. However, it is also possible that ZAP levels are reduced as infection progresses (Figure 6A shows a very clear reduction in ZAPL as infection progresses, but also a more modest reduction in ZAPS). Presumably the antibody used for IF in Figure 6B stains both L and S isoforms? Would the cell-cell variation be diminished in ZAP knockdown cells? Additionally, as ZAP has previously been shown to bind CpG dinucleotides in viral RNA and target it for degradation, it would be interesting if the authors explored the binding of ZAPS to HCMV CpG dinucleotides, or assessed the mRNA levels of the observed ZAPS-antagonized HCMV gene products (e.g. pp52, pp28).

Lines 256-259 and Figure 6B: Contrary to what the authors state, it should be additionally noted that TB40/E-GFP can be certainly used for this microscopy experiment. Rabbit anti-ZAP primary antibody can be detected via an Alexa-fluor-555/568 (or 633/647) conjugated anti-rabbit secondary antibody. Mouse anti-IE1/IE2 primary antibody can be detected via an Alexa-fluor-633/647 (or 555/568) conjugated anti-mouse secondary antibody. This obviates the stated need to switch HCMV strains from TB40/E-GFP to AD169.

In Figure 7, cell sorting gives a population enriched for cells in which infection has progressed the furthest. The loss of ZAPS protein levels over time (Figure 7B) is even clearer in this advanced population. Further supporting the alternate hypothesis, that low ZAPS levels are the result of advanced infection, not the cause.

6) Line 292, and Figure 8A: The authors claim that siTRIM25 results in increased viral proteins levels. This is not true of IE2, or of pp52, although pp28 does appear to be increased. Perhaps the most apparent difference between siTRIM25 and siNEG is the reduced IE1 at days 5/6. Can the authors offer an explanation for this?

7) Figures 8C-F: The authors show that TRIM25 impacts ZAPS protein levels, suggesting that it mediates differential splicing of ZAP. The extent to which ZAP RNA levels are induced (uninfected to infected, or non-treated to IFN treated) is not apparent the way the data is presented. Were these figures generated from a single experiment? If not they should be repeated to show the difference between siNEG and siTRIM25 in relation to the magnitude of induction. Are all of these differences believed to be biologically meaningful (for example, 8C, ZAPS). This approach was considered appropriate in supplemental figure 6, so why not here? Does the n=2 represent full biological replicates? Given the identical exon sequences of ZAPS and ZAPL up to the 699 residues of ZAPS, how are the authors differentiating ZAPS transcript abundance from ZAPL in the “ZAPS” columns? RT-qPCR analysis of any region within ZAPS may also amplify ZAPL.

Reviewer #2: 1. The ultimate experiment to further support the authors’ main conclusion would be to increase the CpG content in the IE1 coding region to test whether this enhances inhibition by ZAP. Ideally, this experiment would be done in the context of the hCMV genome, but that would involve the generation of informative mutant viruses which might be difficult (in part due to potential side effects from mutagenesis within the highly complex major IE region of the viral genome, as mentioned at the end of the Discussion). However, similar experiments could be done using transfected plasmids as in Fig 4. Such experiments would greatly enhance the study.

2. The data on IE2 are less consistent than those on IE1. Although CpG content does not appear to be significantly suppressed in hCMV IE2 (Fig 3), there is little reduction of IE2 (relative to pp52) by ZAP expression in transfected cells (Fig 4). Then again, IE2 levels are quite dramatically reduced by ZAP expression during hCMV infection (Fig 5). The authors should try to explain these somewhat discordant findings.

3. By definition, IE proteins are expressed at very early times during hCMV infection. However, IE2 does not seem to accumulate to significant levels before day three post infection (e.g. Fig 5, 6 and 8), possibly suggesting that the MOI may be lower than specified. Why do the authors think IE2 comes up that late in their system?

Reviewer #3: Fig 4 clearly shows that ZAPS and ZAPL both reduce protein levels of p52 but not IE1, but what about mRNA levels for these viral orfs? Since the authors note in the discussion that previous studies of RNA viruses have suggested distinct mechanisms of action for ZAP, which may or may not include RNA degradation, this should be determined directly for the case of HCMV ie1 vs p52 transcripts. Additionally, as they have shown some very interesting differences in the regulation of ZAPS and ZAPL in various experimental scenarios, these analyses could provide significant insight into how these two differentially spliced transcripts operate.

The authors raise an interesting hypothesis in their discussion that the relative large differences in the levels of CpG throughout the alpha, beta and gamma herpesvirus genomes might be due to latency establishment in distinct cell types. It is quite likely that CMV establishes latency in endothelial cells in addition to hematopoietic cells, and little evidence suggest fibroblasts are true latent reservoirs. As the authors have performed the vast majority of their studies with the TB40/E strain, this provides them an opportunity to assess whether ZAP/TRIM25 function in this cell type to suppress non-IE gene expression similar to fibroblasts. So many questions exist in this field regarding whether a particular host innate defense mechanism operates similarly in distinct cell types. Even just assessing whether ZAP S and L are expressed in human endothelial cells (e.g. HUVEC), are inducible via IFN-I signaling and if their overexpression shows similar inhibition of HCMV replication would increase the impact of their study significantly.

**Part III – Minor Issues: Editorial and Data Presentation Modifications**

Reviewer #1: 1) Introduction: Sentence beginning on line 68 (“RNA and small DNA…”) should be split into 2 sentences for clarity.

2) Line 153: “dependent” is misspelled.

3) Line 170: “efficient cGAS inhibition” could have mixed interpretations so would suggest re-wording.

4) Line 294-297: Please cite the literature.

Reviewer #2: 4. The title and parts of the main text (e.g. lines 200-201) suggest that there is more than one major IE gene and that all of them are affected by CpG suppression, which does not seem to be the case.

5. I do not agree based on the presented data that “ZAP and TRIM25 are rapidly reduced” during hCMV infection (line 34). They are reduced, but not rapidly and maybe not efficiently enough to explain the normal CpG content in viral genes other than IE1. Can the authors speculate about hCMV antagonists of ZAP/TRIM25?

6. Combine sentence in lines 90-91 with previous paragraph.

7. Provide reference for “Individuals with mutations in key IFN signalling genes are lethally susceptible to HCMV infections…” (lines 102-103).

8. Clarify that IRF3, IRF7 and NFKB typically activate IFN genes first, and that ISGs are usually activated as a consequence of that (line 112).

9. Specify (in figure legend) or remove horizontal red line in Fig 1D.

10. Correct “These genes are the first viral transcripts to be expressed…” (line 201), since genes are not transcripts.

11. Remove full stops from subheadings (line 211 and 244).

12. All abbreviations used in the figures, including “EMP” and “UN”, should be defined in the figure legends.

13. Discuss whether CpG suppression extends to the hCMV major IE promoter-enhancer (and other non-coding sequences) or if it is restricted to the open reading frames. What could this mean, if CpG suppression is not limited to open reading frames?

14. Replace “student T test” (e.g. legend to Fig 8) with “Student’s t-test”.

15. Reference 47 is about the role of ZAP in alphavirus, not alpha-herpesvirus (HSV-1) replication (line 378-380). Might the fact that alpha-herpesviruses replicate more quickly (express ZAP/IFN antagonists earlier?) than beta-herpesviruses explain partly the differences in CpG suppression?

Reviewer #3: The data in Fig 6 showing that at the individual cell level, cells expressing higher levels of ZAPS show reduced IE2 expression, but not IE1, is a very elegant result.

Data shown in Fig. 8 indicating that TRIM25 can regulate the differential splicing of ZAP between its long and short isoforms is the first example of TRIM family proteins operating at this level, which is highly interesting.

The paper is very clearly written, it was an enjoyable read.

PLOS authors have the option to publish the peer review history of their article (what does this mean?). If published, this will include your full peer review and any attached files.

Reviewer #1: No

Reviewer #2: Yes: Michael M Nevels

Reviewer #3: No
---

## [Editor Report · Decision Letter 1]

28 Jul 2020

Dear Career Track Fellow Grey,

We are pleased to inform you that your manuscript 'Human cytomegalovirus evades ZAP detection by suppressing CpG dinucleotides in the major immediate early 1 gene' has been provisionally accepted for publication in PLOS Pathogens.

Best regards,

Eain A Murphy, Ph.D.

Associate Editor

PLOS Pathogens

Blossom Damania

Section Editor

PLOS Pathogens

Kasturi Haldar

Editor-in-Chief

PLOS Pathogens

orcid.org/0000-0001-5065-158X

Michael Malim

Editor-in-Chief

PLOS Pathogens

orcid.org/0000-0002-7699-2064

Dr Grey.

I hope this finds you well. I have looked over the revised manuscript with a particular focus on the response to previous review and believe you have addressed the concerns of the three expert reviewers sufficiently to warrant acceptance of your manuscript in its present form. Congratulations. It is a nice piece of work and you should be proud of it.

Sincerely,

Eain Murphy
---

## [Editor Report · Acceptance letter]

31 Aug 2020

Dear Career Track Fellow Grey,

We are delighted to inform you that your manuscript, "Human cytomegalovirus evades ZAP detection by suppressing CpG dinucleotides in the major immediate early 1 gene," has been formally accepted for publication in PLOS Pathogens.

Best regards,

Kasturi Haldar

Editor-in-Chief

PLOS Pathogens

orcid.org/0000-0001-5065-158X

Michael Malim

Editor-in-Chief

PLOS Pathogens

orcid.org/0000-0002-7699-2064